# Metasurfaces and Blinking Jamming: Convergent Study, Comparative Analysis, and Challenges

**DOI:** 10.3390/mi14071405

**Published:** 2023-07-11

**Authors:** Rafael Gonçalves Licursi de Mello

**Affiliations:** Greenerwave, 75002 Paris, France; rafael.licursi@ieee.org

**Keywords:** blinking jamming, monopulse radar, electronic warfare, radar cross-section (RCS), metasurface absorbers, power-amplifying metasurfaces, reconfigurable metasurfaces

## Abstract

Blinking jamming is an active self-screening technique performed by at least two aircraft to tackle monopulse radars and all complexity related thereto. Nowadays, the technique can be performed with digital radiofrequency memories (DRFMs), which are cumbersome, complex, expensive, need a dedicated compartment and antenna, and introduce spurs in the signals. In this paper, we propose an alternative to the implementation of blinking jamming with DRFMs, namely with reconfigurable metasurfaces. By covering the aircraft parts that most contribute to the radar cross-section (RCS), reconfigurable metasurfaces can interchangeably absorb or amplify impinging waves, making the aircraft ‘blink’ from the radar perspective. To validate the feasibility, simulations accounting for realistic phenomena are conducted. It is seen that, if the aircraft RCS can be varied in a ratio of 10:1, either with absorptive or power-amplifying metasurfaces, a performance similar to that of the DRFM is achieved. Furthermore, a ratio of 2:1 is sufficient to make the radar antenna system movements exceed the angular range of the formation. We also anticipate our work to be a starting point for completely new ways of countering radars, e.g., with countless small drones performing passive or active stand-off blinking jamming.

## 1. Introduction

The nature of electronic warfare (EW), based on the adoption of measures and countermeasures to achieve dominance of the electromagnetic spectrum, makes it an inherently dynamic field where the rate of innovation plays a central role. Since its beginnings, EW has been characterized by opponents deploying new technologies to counter one another in a perpetual cycle that has pushed the state-of-the-art in many ways. The radar technology, in particular, evolved from primitive detection systems to conical-scan and monopulse tracking radars in a fast pace [1].

The conical-scan radar represented a watershed in defense because it allowed an unprecedentedly accurate tracking of hostile platforms. However, such radar was highly susceptible to electronic attack (EA) measures, namely jamming signals emulating false angle information about the target. The main reason for this vulnerability is that conical-scan radars use time-series information contained in received signals in order to resolve for the angular position of targets. When a target is not in the center of the conical scan, an amplitude modulation appears in the envelope of the time series according to its displacement. Angle deception jamming techniques can mimic this modulation [2].

In this context, it is fair to say that the arrival of the monopulse radar represented a remarkable challenge to EW engineers. Such radar, illustrated in Figure 1, uses four antennas, A–D, sharing a common reflector. In the receiving, the pulse returns can be reasonably discriminated in four receivers, A–D. As such, the radar can resolve the angular position of targets with a single pulse return rather than using time-series information. For this reason, monopulse radars are robust compared to the jamming techniques of the early days. More than that, conventional jammers can even work as beacons, eventually putting their own platforms at risk when facing monopulse radars that guide weapons [3].

Blinking jamming, an active self-screening jamming technique whose potential was envisioned back in the 1950s [4], is one of the solutions to tackle monopulse radars. The technique uses a formation of two or more aircraft that alternately transmit jamming signals over the radar. The platforms should perform a specific flight path so as to be angularly separated in relation to the target radar, but within the same resolution cell. As such, the radar tracking jumps from one platform to another, and, in the end, the guided weapon misses them all. Although high-level models of blinking jamming have been presented in the open literature for decades [5,6,7], it was not until 2020 that models detailed enough to provide scientific insight to improve the technique were proposed [8,9]. In [8], a theoretical analysis is introduced, taking into account the ratio between the jamming signals and the skin returns in each channel of the monopulse radar, leading to a new amplitude-modulated blinking jamming. In [9], an analysis on stealth formations using blinking jamming at different conditions (diverse speeds, releasing projectiles or not) is presented.

The analyses on blinking jamming presented in [5,6,7,8,9] consider that the jamming signals reach the heart of the radar processing without being attenuated by the radar filters. Such a scenario is fairly realistic nowadays if the technique is implemented through the use of digital radiofrequency memories (DRFMs) [10]. A DRFM is an equipment that uses an antenna and a radiofrequency (RF) front-end to intercept radar signals, and an analog-to-digital conversion (ADC) circuit to digitize and store them. Then, a processor may introduce slight modifications to the signals for deception purposes, and a digital-to-analog conversion (DAC) circuit converts them from the digital to the analog domain. Last, another RF front-end amplifies the copied/modified signals before retransmission. Because the jamming signal is close to the original one, two significant positive points exist in this approach: First, the jamming is efficient in terms of power/bandwidth; that is, the power is concentrated over the frequencies of the original signal. Second, the jamming signal is normally accepted by coherent radars, being eventually amplified by pulse compression gain stages, like the original signal was [11].

Although DRFMs are considered one of the most advanced EA technologies today, some limitations are inherent to it. Due to its digital nature, undesired spurs may appear in the signals that DRFMs reproduce, whereby triggering electronic protection (EP) measures in advanced radar systems [12]. It is reported in [13] that 2-bit DRFMs typically produce spurs about −18.1 dB below the peak, and 4-bit DRFMs produce about −34.4 dB. Moreover, the DRFM hardware is complex and expensive. Its power consumption, bulkiness, and need for an antenna and dedicated compartment keep DRFMs from being deployed, for example, in small drones.

Two main solutions exist for the problem of spurious components in the jamming signals. The first is to increase the number of bits of the ADC and DAC circuits in the DRFM, which has an impact in the power consumption, computational workload, and ultimately in the bulkiness, complexity, and cost of the equipment [14]. The second is to replace the digital technique with photonic approaches, which also have a high degree of complexity [15,16]. Neither solution mitigates the bulkiness inconvenience associated with the need of an antenna and dedicated compartment or the power consumption problem related to the retransmission of signals.

In this paper, we propose another alternative to the digital approach in blinking jamming, namely by using reconfigurable metasurfaces. Metasurfaces are artificially engineered structures composed of polarizable unit cells. The subwavelength dimensions of the unit cells allow the structure to be treated as an effective surface impedance [17]. Moreover, their resonating nature allows one to obtain arbitrary reflection, transmission, and absorption coefficients from each cell [18]. As such, metasurfaces enable the arbitrary shaping of electromagnetic waves. Figure 2 exemplifies the physical disposition of a metasurface known as an artificial magnetic conductor (AMC), whose unit cell is a square patch spaced by a dielectric, hidden for the sake of visualization, from a ground plane.

Metasurfaces can be implemented by etching, drilling, and deploying lumped elements over dielectric laminates, exactly as performed in conventional printed circuit boards. Among the capabilities unlocked by metasurfaces are the absorption [19,20,21] and control of the polarization [22,23] of waves; the steering [24,25], focusing [18,26], and shaping [18,24,25,26,27] of beams; and the design of multiband reflectors [28,29] and reflectors with a power amplification [30,31]. The unusual properties of these engineered surfaces have been thoroughly studied in the last couple of decades [32]. In the beginning, such properties were considered controversial, but now they are acknowledged by academia.

To the best of the author’s knowledge, this is the first study envisioning the application of metasurfaces in blinking jamming. Specifically, two types of blinking jamming are considered: passive and active. The importance of this work lies in the fact that metasurfaces are effectively reaching real applications in the market today [33] and, considering the innovative nature of EW, this field cannot be put aside. The purpose is to show that a development of this technology can lead to the following advantages in relation to DRFMs or photonic approaches:A passive blinking jamming performed with reconfigurable metasurface absorbers mitigates the power consumption associated with a retransmission of signals;An active blinking jamming performed with power-amplifying metasurfaces is free of spurs due to the non-digital nature of metasurfaces which avoids the trigger of EP measures in advanced radar systems;For both passive and active blinking jamming with metasurfaces covering strategic parts of the platform, the need for a dedicated compartment and an antenna to intercept and retransmit signals is eliminated.

The use of metasurfaces can lead not only to performances better than that of the DRFM, but also to new ways of performing EA, e.g., using small drones to conduct stand-off blinking jamming operations on a large scale in the theater of operation.

Here, we present first a study on blinking jamming as it is performed today, making clear the advantages and limitations of the technique when applied against monopulse radars. Next, the paper provides a general introduction on metasurfaces and review candidate structures that could be used in blinking jamming. For each technique (passive and active), one metasurface is chosen in the literature, and simulations are conducted while taking into account their measured performances. For comparison purposes, we also simulate a scenario considering the performance of an ideal DRFM. As such, the benefits and limitations of the use of metasurfaces in blinking jamming become clear. Lastly, the challenges to overcome before metasurfaces can indeed reach the theater of operation in this application are also discussed.

## 2. Materials and Methods

### 2.1. Problem Formulation: The Monopulse Radar

Consider that the monopulse radar of Figure 1 transmits identical signals through four channels, i.e., A–D. Each channel uses a single feed antenna symmetrically squinted by $\theta_{sqt}$ and $\Phi_{sqt}$ in relation to the broadside direction of a shared reflector, constituting together the radar antenna system, which, in turn, is pointed at the direction $\theta_{ant}$ and $\Phi_{ant}$. For simplicity, we consider in this work that the mutual coupling between the feed antennas is negligible. Therefore, the radiation pattern of the antenna system presents four identical lobes squinted in relation to the broadside direction of the antenna system, identified in Figure 1 as lobes A–D. Moreover, their signals can be reasonably discriminated from one another in the respective receivers A–D.

The isolate radiation patterns, $G_{A-B}\left(\theta ,\Phi \right)$, associated with each channel (i.e., A–D) separately can be denoted in terms of a reference radiation pattern, $G_{0}\left(\theta ,\Phi \right)$, squinted in relation to the broadside direction, as follows:(1)$$G_{A}\left(\theta ,\Phi \right)=G_{0}\left(\theta -\theta_{sqt}-\theta_{ant},\Phi +\Phi_{sqt}-\Phi_{ant}\right)$$
(2)$$G_{B}\left(\theta ,\Phi \right)=G_{0}\left(\theta +\theta_{sqt}-\theta_{ant},\Phi +\Phi_{sqt}-\Phi_{ant}\right)$$
(3)$$G_{C}\left(\theta ,\Phi \right)=G_{0}\left(\theta -\theta_{sqt}-\theta_{ant},\Phi -\Phi_{sqt}-\Phi_{ant}\right)$$
(4)$$G_{D}\left(\theta ,\Phi \right)=G_{0}\left(\theta +\theta_{sqt}-\theta_{ant},\Phi -\Phi_{sqt}-\Phi_{ant}\right)$$

Now, consider an aircraft flying with speed $v_{1}$ being illuminated by the monopulse radar. The radar is at the origin of the cartesian coordinate system denoted in Figure 1, that is, $P\left(0,0,0\right)$. The aircraft is in a radius, $r_{acf}$, at the azimuthal angle, $\Phi_{acf}$, and elevation angle, $\theta_{acf}$, that is, in the position $P\left(r_{acf},\theta_{acf},\Phi_{acf}\right)$ of the spherical coordinate system, also indicated in Figure 1. Figure 3a illustrates this scenario in the azimuth plane, whereas Figure 3b does so in the elevation plane.

If the radar is pointing the broadside of its antenna system to the aircraft, that is, $\theta_{ant}=\theta_{acf}$ and $\Phi_{ant}=\Phi_{acf}$, then the skin return, $S_{A-D}$, of the aircraft in each channel, i.e., A–D, can be denoted as follows:(5)$$\begin{array}{cc} S_{A} & =\frac{P_{t}\lambda^{2}}{{\left(4\pi \right)}^{3}r_{acf}^{4}}G_{A}^{2}\left(\theta =\theta_{acf},\Phi =\Phi_{acf}\right)\sigma \left({\theta ’}_{rdr},{\Phi ’}_{rdr}\right)\\ & =\frac{P_{t}\lambda^{2}}{{\left(4\pi \right)}^{3}r_{acf}^{4}}G_{0}^{2}\left(-\theta_{sqt},+\Phi_{sqt}\right)\sigma \left({\theta ’}_{rdr},{\Phi ’}_{rdr}\right) \end{array}$$
(6)$$\begin{array}{cc} S_{B} & =\frac{P_{t}\lambda^{2}}{{\left(4\pi \right)}^{3}r_{acf}^{4}}G_{B}^{2}\left(\theta =\theta_{acf},\Phi =\Phi_{acf}\right)\sigma \left({\theta ’}_{rdr},{\Phi ’}_{rdr}\right)\\ & =\frac{P_{t}\lambda^{2}}{{\left(4\pi \right)}^{3}r_{acf}^{4}}G_{0}^{2}\left(+\theta_{sqt},+\Phi_{sqt}\right)\sigma \left({\theta ’}_{rdr},{\Phi ’}_{rdr}\right) \end{array}$$
(7)$$\begin{array}{cc} S_{C} & =\frac{P_{t}\lambda^{2}}{{\left(4\pi \right)}^{3}r_{acf}^{4}}G_{C}^{2}\left(\theta =\theta_{acf},\Phi =\Phi_{acf}\right)\sigma \left({\theta ’}_{rdr},{\Phi ’}_{rdr}\right)\\ & =\frac{P_{t}\lambda^{2}}{{\left(4\pi \right)}^{3}r_{acf}^{4}}G_{0}^{2}\left(-\theta_{sqt},-\Phi_{sqt}\right)\sigma \left({\theta ’}_{rdr},{\Phi ’}_{rdr}\right) \end{array}$$
(8)$$\begin{array}{cc} S_{D} & =\frac{P_{t}\lambda^{2}}{{\left(4\pi \right)}^{3}r_{acf}^{4}}G_{D}^{2}\left(\theta =\theta_{acf},\Phi =\Phi_{acf}\right)\sigma \left({\theta ’}_{rdr},{\Phi ’}_{rdr}\right)\\ & =\frac{P_{t}\lambda^{2}}{{\left(4\pi \right)}^{3}r_{acf}^{4}}G_{0}^{2}\left(+\theta_{sqt},-\Phi_{sqt}\right)\sigma \left({\theta ’}_{rdr},{\Phi ’}_{rdr}\right) \end{array}$$
where $P_{t}$ is the power transmitted through each channel, A–D; λ is the wavelength; σ is the monostatic radar cross-section (RCS) of the aircraft; ${\theta ’}_{rdr}$ is the radar elevation angle; and ${\Phi ’}_{rdr}$ is its azimuthal angle, which refers to the aircraft orientation axes.

As we can see in (5)–(8), the skin returns, $S_{A-D}$, of the aircraft in each channel (A–D) depend on the squinted radiation patterns, $G_{A-D}\left(\theta ,\Phi \right)$. Because of that, an amplitude-comparison method can be performed to estimate the angles $\theta_{acf}$ and $\Phi_{acf}$ of the aircraft whenever they diverge from $\theta_{ant}$ and $\Phi_{ant}$. To this end, the monopulse radar uses the actual channels A–D to create three virtual channels as follows:(9)$$S_{\sum }=S_{A}+S_{B}+S_{C}+S_{D}$$
(10)$$S_{∆\Phi }=\left(S_{A}+S_{B}\right)-\left(S_{C}+S_{D}\right)$$
(11)$$S_{∆\theta }=\left(S_{A}+S_{C}\right)-\left(S_{B}+S_{D}\right)$$
where $S_{\sum }$ is known as the sum channel; and $S_{∆\Phi }$ and $S_{∆\theta }$ are, respectively, the azimuthal and elevation difference channels. The angular error of the target in relation to the broadside direction of the radar antenna system can be calculated next as follows:(12)$$\Phi_{\epsilon }=k_{\Phi }\frac{S_{∆\Phi }}{S_{\sum }}=\Phi_{acf}-\Phi_{ant}$$
(13)$$\theta_{\epsilon }=k_{\theta }\frac{S_{∆\theta }}{S_{\sum }}=\theta_{acf}-\theta_{ant}$$
where $\Phi_{\epsilon }$ and $\theta_{\epsilon }$ are, respectively, the azimuth and elevation errors; and $k_{\Phi }$ and $k_{\theta }$ are the proportionality constants mapping the sensitiveness of the normalized amplitudes, $S_{∆\Phi }/S_{\sum }$ and $S_{∆\theta }/S_{\sum }$, to such errors.

The azimuth and elevation errors, $\Phi_{\epsilon }$ and $\theta_{\epsilon }$, are used to feed the radar antenna base control system, which steers the antenna system to keep the aircraft at its broadside direction. For the sake of simplicity, in this work, we model the antenna base control system as two independent servos whose behaviors are similar to a simple mass–spring–damper system. The servos make the orientation of the antenna system, $\Phi_{ant}$ and $\theta_{ant}$, converge to the aircraft angular position, $\theta_{acf}$ and $\Phi_{acf}$, thereby minimizing the $\Phi_{\epsilon }$ and $\theta_{\epsilon }$:(14)$$\underset{\begin{array}{c} driving\\ force \end{array}}{\underbrace{\Phi_{\epsilon }}}=\underset{inertia}{\underbrace{m_{ant}\frac{\partial^{2}\Phi_{ant}}{\partial t^{2}}}}+\underset{friction}{\underbrace{m_{ant}d_{\Phi }\frac{\partial \Phi_{ant}}{\partial t}}}-\underset{\begin{array}{c} restoring\\ force \end{array}}{\underbrace{m_{ant}\omega_{\Phi }^{2}\Phi_{\epsilon }}}$$
(15)$$\underset{\begin{array}{c} driving\\ force \end{array}}{\underbrace{\theta_{\epsilon }}}=\underset{inertia}{\underbrace{m_{ant}\frac{\partial^{2}\theta_{ant}}{\partial t^{2}}}}+\underset{friction}{\underbrace{m_{ant}d_{\theta }\frac{\partial \theta_{ant}}{\partial t}}}-\underset{\begin{array}{c} restoring\\ force \end{array}}{\underbrace{m_{ant}\omega_{\theta }^{2}\theta_{\epsilon }}}$$
where $m_{ant}$ is an equivalent mass of the antenna system; $d_{\Phi }$ and $d_{\theta }$ are damping coefficients; and $\omega_{\Phi }=2\pi f_{\Phi }$ and $\omega_{\theta }=2\pi f_{\theta }$ are the angular natural frequencies of the system in the azimuth and elevation dimensions, respectively.

We exemplify the behavior of the antenna control system represented by (14) and (15) by depicting, in Figure 4, the angular position, $\Phi_{ant}$ and $\theta_{ant}$, when the monopulse radar, initially oriented to $\Phi_{ant}=0{}^{\circ}$ and $\theta_{ant}=0{}^{\circ}$, detects an aircraft at $\Phi_{acf}=5{}^{\circ}$ and $\theta_{acf}=1.5{}^{\circ}$. The antenna equivalent mass is $m_{ant}=10\;kg$; the damping coefficients in the azimuthal and elevation dimensions are, respectively, $d_{\Phi }=10/s$ and $d_{\theta }=6/s$; and the natural frequencies are $f_{\Phi }=4\;Hz$ and $f_{\theta }=2\;Hz$.

Note in Figure 4 how the antenna system converges to the aircraft angular position. Some oscillations are initially seen, in accordance with the natural frequencies in both the azimuthal and elevation dimensions. Such oscillations are quickly attenuated by the damping coefficients in the respective dimensions.

### 2.2. Blinking Jamming

Now consider a dual-aircraft formation being illuminated by the same monopulse radar previously stated. The formation is tight enough to keep both Aircraft 1 and 2 in an azimuthal angular range of $∆\Phi <\Phi_{3dB}$, as depicted in Figure 5a. Likewise, Aircraft 1 and 2 lie in the elevation angular range of $∆\theta <\theta_{3dB}$, as indicated in Figure 5b. Here, $\Phi_{3dB}$ and $\theta_{3dB}$ are the 3 dB beamwidths of the virtual lobe in the sum channel, $S_{\sum }$, in the azimuthal and elevation planes, respectively. Furthermore, the difference of ranges of Aircraft 1 and 2 in relation to the radar is $∆r<c_{0}/2BW$, where $c_{0}$ is the speed of the light, and BW is the radar bandwidth. Their speeds are $v_{1}=v_{2}$. Putting it differently, both aircraft lie in the same radar resolution cell.

If Aircraft 1 and 2 in Figure 5 were each illuminated alone by the four antenna lobes, different angular error sets, [$\theta_{\epsilon 1}$, $\Phi_{\epsilon 1}$] and [$\theta_{\epsilon 2}$, $\Phi_{\epsilon 2}$], would be estimate, respectively, by the radar according to their displacements, and, from (12) and (13), the signals in the difference channels would be as follows:(16)$$S_{∆\Phi 1}=\frac{\Phi_{\epsilon 1}S_{\sum 1}}{k_{\Phi }},{\;S}_{∆\Phi 2}=\frac{\Phi_{\epsilon 2}S_{\sum 2}}{k_{\Phi }}$$
(17)$$S_{∆\theta 1}=\frac{\theta_{\epsilon 1}S_{\sum 1}}{k_{\theta }},\;S_{∆\theta 2}=\frac{\theta_{\epsilon 2}S_{\sum 2}}{k_{\theta }}$$

However, since the aircraft are in the same resolution cell, their skin returns will be superimposed in the radar receivers, that is, in (12) and (13), $S_{\sum }=S_{\sum 1}+S_{\sum 2}$, $S_{∆\Phi }=S_{∆\Phi 1}+S_{∆\Phi 2}$, and $S_{∆\theta }=S_{∆\theta 1}+S_{∆\theta 2}$. Hence, using (16) and (17), the angle errors estimated by the monopulse radar is calculated as follows:(18)$$\Phi_{\epsilon }=\frac{\Phi_{\epsilon 1}S_{\sum 1}+\Phi_{\epsilon 2}S_{\sum 2}}{S_{\sum 1}+S_{\sum 2}}$$
(19)$$\theta_{\epsilon }=\frac{\theta_{\epsilon 1}S_{\sum 1}+\theta_{\epsilon 2}S_{\sum 2}}{S_{\sum 1}+S_{\sum 2}}$$

Equations (18) and (19) show that the estimated angle errors, $\Phi_{\epsilon }$ and $\theta_{\epsilon }$, are the sum of the skin returns of Aircraft 1 and 2 in the sum channel, respectively, weighted by the angle displacements of the aircraft in relation to the broadside direction of the antenna system [8]. Only the skin returns are considered in (18) and (19).

Now, suppose that Aircraft 1 is capable of transmitting jamming signals, which are, respectively, received by the channels A–D of the monopulse radar as follows:(20)$$J_{A-D}=\frac{P_{j}\lambda^{2}}{{\left(4\pi r_{acf}\right)}^{2}}G_{j}\left({\theta ’}_{rdr},{\Phi ’}_{rdr}\right)G_{A-D}\left(\theta_{acf}-\theta_{ant},\Phi_{act}-\Phi_{ant}\right)$$
where $J_{A-D}$ are the jamming signals seen in each channel, i.e., A–D; $P_{j}$ is the power transmitted by the jammer; and $G_{j}\left({\theta ’}_{rdr},{\Phi ’}_{rdr}\right)$ is the gain of the jammer antenna referred to the orientation axes of Aircraft 1. Consider that these signals suffer attenuations and amplifications identical to those of the skin returns in the radar processing. As such, the jamming signals $J_{A-D}$ are superimposed to the skin returns $S_{A-D}$ in the radar channels. Therefore, we can consider that the signals $J_{A-D}$ appear as a signal $J_{1}$ in the sum virtual channel. In this case, (18) and (19) become the following:(21)$$\Phi_{\epsilon }=\frac{\Phi_{\epsilon 1}\left(S_{\sum 1}+J_{1}\right)+\Phi_{\epsilon 2}S_{\sum 2}}{S_{\sum 1}+S_{\sum 2}+J_{1}}$$
(22)$$\theta_{\epsilon }=\frac{\theta_{\epsilon 1}\left(S_{\sum 1}+J_{1}\right)+\theta_{\epsilon 2}S_{\sum 2}}{S_{\sum 1}+S_{\sum 2}+J_{1}}$$

From (21) and (22), we can see that $\Phi_{\epsilon }\to \Phi_{\epsilon 1}$ and $\theta_{\epsilon }\to \theta_{\epsilon 1}$ when $J_{1}\gg S_{\sum 2}$. Analogously, if Aircraft 2 is capable of transmitting jamming signals $J_{2}$, then it can make $\Phi_{\epsilon }\to \Phi_{\epsilon 2}$ and $\theta_{\epsilon }\to \theta_{\epsilon 2}$ when $J_{2}\gg S_{\sum 1}$ and Aircraft 1 is not transmitting jamming signals ($J_{1}=0$).

From the above development, we can see that DRFMs retransmitting the same waveforms employed by the radar can make $\Phi_{\epsilon }\to \Phi_{\epsilon 1},\Phi_{\epsilon 2}$ and $\theta_{\epsilon }\to \theta_{\epsilon 1},\theta_{\epsilon 2}$. This is performed with the introduction of the high-power jamming signals $J_{1}\;and\;J_{2}$ through the coherent processing of the radar. One can take advantage of these facts to make the radar antenna control system oscillate between the angles $\Phi_{\epsilon 1}$ and $\Phi_{\epsilon 2}$, and/or $\theta_{\epsilon 1}$ and $\theta_{\epsilon 2}$. By alternately transmitting jamming signals $J_{1}$ and $J_{2}$ over the target radar with a blinking frequency, $f_{bl}$, the natural frequencies $f_{\Phi }$ and $f_{\theta }$ of the control system can be exploited. As such, the oscillations can even exceed $\Phi_{\epsilon 1}$ and $\Phi_{\epsilon 2}$, and/or $\theta_{\epsilon 1}$ and $\theta_{\epsilon 2}$, and the guidance of weapons by the target radar will bounce around the jamming platforms, becoming ineffective.

We exemplify such a situation by depicting, again, the responses of the antenna control system represented by (14) and (15), considering an initial orientation of $\Phi_{ant}=7{}^{\circ}$ and $\theta_{ant}=12{}^{\circ}$. The antenna equivalent mass, the damping coefficients, and the natural frequencies are equal to the ones considered in Figure 4 ($m_{ant}=10\;kg$, $d_{\Phi }=10/s$, $d_{\theta }=6/s$, $f_{\Phi }=4\;Hz$, and $f_{\theta }=2\;Hz$). The radar detects an aircraft at $\Phi_{acf}=10{}^{\circ}$ and $\theta_{acf}=15{}^{\circ}$, which actually is a dual-aircraft formation. The aircraft are azimuthally separated by ∆Φ=1°. On the other hand, the separation in the elevation plane is ∆θ=0.5°. The jamming is activated from the instant t=1.5 s. It is assumed here that the error information input to the antenna system is precise (we simply use $\Phi_{\epsilon }=\Phi_{acf}-\Phi_{ant}$ and $\theta_{\epsilon }=\theta_{acf}-\theta_{ant}$). Figure 6a shows the radar antenna angular position $\Phi_{ant}$ and $\theta_{ant}$ (continuous blue and dashed red lines) when the blinking frequency, $f_{bl}$, matches the natural frequency, $f_{\theta }$, of the radar antenna control system in the elevation dimension. The aircraft that is effectively transmitting the jamming signals at each instant is also indicated (dotted black line).

We can see in Figure 6a that, from the instant t=1.5 s, the radar cannot stable its pointing towards the formation anymore. The matching between the blinking frequency, $f_{bl}$, and the radar control system natural frequency in elevation, $f_{\theta }$, causes significant oscillations in this dimension, ranging from 14.33° to 15.66° and exceeding the angular position of both aircraft. In azimuth, the oscillations range from 9.09° to 10.91°, also exceeding the aircraft separation. A weapon guided by such radar would probably miss both aircraft.

In Figure 6b, we depict the system response when the blinking frequency, $f_{bl}$, matches the natural frequency, $f_{\Phi }$, in azimuth, and significant oscillations occur at this dimension, ranging from 8.39° to 11.61°. In the elevation dimension, the oscillations decrease, going from 14.90° to 15.10°. Nevertheless, a weapon guided by such radar would probably miss both aircraft since the error in one dimension is sufficient for that.

Lastly, Figure 6c illustrates the system response when the blinking frequency is the average of the natural frequencies in both azimuthal and elevation dimensions. A good compromise is found, with a range from 8.81° to 11.19° in the azimuthal dimension and 14.77° to 15.22° in the elevation dimension.

**Figure 6 micromachines-14-01405-f006:**
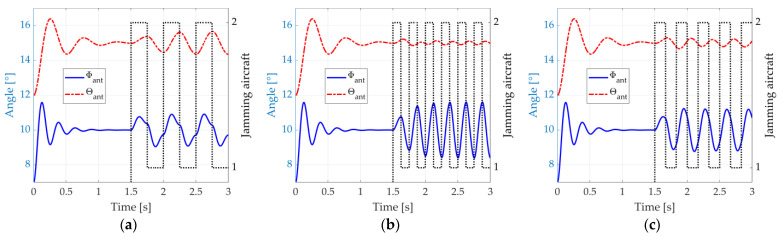
Response of the antenna control system to the dual-aircraft formation, suddenly detected at the angular position $\Phi_{acf}=5{}^{\circ}$ and $\theta_{acf}=1.5{}^{\circ}$. The jamming starts from t=1.5 s : (**a**) $f_{bl}=f_{\theta }$, (**b**) $f_{bl}=f_{\Phi }$, and (**c**) $f_{bl}=\left(f_{\Phi }+f_{\theta }\right)/2$.

The above responses show the potential of blinking jamming. As already mentioned, achieving such a response with DRFMs involves installing in each aircraft an antenna and an RF front end to receive impinging signals, a complex circuit to digitize and store them, another circuit to convert them from the digital to the analog domain, and another RF front end for retransmission with power amplification. Moreover, the creation of spurs and a high-power consumption are expected. However, we can see in (18) and (19) that another way of making $\Phi_{\epsilon }\to \Phi_{\epsilon 1}$ and $\theta_{\epsilon }\to \theta_{\epsilon 1}$ is by making $S_{\sum 2}\ll S_{\sum 1}$. Similarly, $\Phi_{\epsilon }\to \Phi_{\epsilon 2}$ and $\theta_{\epsilon }\to \theta_{\epsilon 2}$ when $S_{\sum 1}\ll S_{\sum 2}$.

### 2.3. A Brief Introduction to Metasurfaces

As previously stated, metasurfaces are artificially engineered structures composed of polarizable unit cells whose subwavelength dimensions allow the structure to be treated as an effective surface impedance [17]. The work that popularized metasurfaces [34] was about a resonating structure that could mimic the behavior of a perfect magnetic conductor (PMC) at a specific frequency. A surface made of a perfect electric conductor (PEC) provides a reflection coefficient of Γ=−1 for impinging electromagnetic waves; that is, it reflects all the energy of these waves with a phase shift of $\varphi_{\Gamma }=180{}^{\circ}$. On the other hand, the PMC is a theoretical material whose reflection coefficient is Γ=+1. Thus, it reflects all the energy of the waves but with a phase shift of $\varphi_{\Gamma }=0{}^{\circ}$.

Although such a property finds many applications in microwave devices, PMCs do not exist in nature. In [34], a textured surface composed of periodic metal patches connected through metal vias to a ground plane was proposed to emulate this behavior at a specific frequency. For this reason, the structure is considered an AMC. Other types of AMCs exist [35,36] with different shapes for the metal patches, which are isolated from the ground plane by a dielectric. These structures may have vias or not. The metasurface earlier exemplified in Figure 2 is an AMC without vias.

The layers that compose a metasurface can be modelled as surface impedances representing the ratio between the electric and magnetic fields thereon. As such, a transmission-line equivalence may be drawn. For instance, the AMC of Figure 2 can be seen as a frequency selective surface (FSS) of impedance, $Z_{FSS}$ (the patches), over a grounded slab. The grounded slab, made of a dielectric with permeability, $\mu_{diel}$; permittivity, $\varepsilon_{diel}$; and thickness, $l_{diel}$, over a PEC ground plane, can be seen as a transmission line. This line has a characteristic impedance of $Z_{diel}=\sqrt{\mu_{diel}/\varepsilon_{diel}}$ and length of $l_{diel}$. It leads to a short circuit, whose impedance is $Z_{short}=0$. Figure 7 depicts the transmission-line equivalent of such an AMC.

The input impedance, $Z_{slab}$, of the grounded slab indicated in Figure 7 can be calculated using the conventional formulation for a short-circuit stub in transmission lines:(23)$$Z_{slab}=jZ_{diel}\tan\left(k_{slab}l_{diel}\right)$$
where $k_{slab}=\omega \sqrt{\mu_{diel}\varepsilon_{diel}}$ is the wavenumber within the dielectric. Note that, in the case of $l_{diel}\ll \lambda$, we have the following:(24)$$\begin{array}{cc} \underset{l\to 0}{\lim}\left(Z_{slab}\right)= & jZ_{diel}k_{slab}l_{diel}\\ Z_{slab}≅ & j\omega \mu_{diel}l_{diel}=j\omega L_{slab} \end{array}$$
where $L_{slab}=\mu_{diel}l_{diel}$ is an inductance that we can assign to the grounded slab.

The surface impedance, $Z_{s}$, of the AMC at the reference plane of the FSS is the parallel of $Z_{slab}$ and $Z_{FSS}$. The surface impedance, $Z_{FSS}$, can be estimated by simulating or measuring the reflection coefficient of the FSS alone when illuminated by a normally incident plane wave. For the specific case where the impedance of the standalone FSS is purely capacitive (i.e., we can represent it by $Z_{FSS}=1/j\omega C_{FSS}$), we have the following:(25)$$Z_{s}=j\frac{\omega L_{slab}}{1-\omega^{2}L_{slab}C_{FSS}}$$

We can see that, when $\omega =1/\sqrt{L_{slab}C_{FSS}}$, $Z_{s}=j\infty$. Therefore, a reflection coefficient phase of $\varphi_{\Gamma }=0{}^{\circ}$ is provided.

The transmission-line equivalence is also useful for the modelling of other types of metasurfaces. For instance, in [37], a metasurface absorber was proposed by adding a resistive sheet on the top of an AMC. Using an equivalent transmission line like the one in Figure 7, the surface impedance, $Z_{s}$, now is the parallel of the AMC impedance, $Z_{AMC}$, and the impedance, $R_{sheet}$, of the resistive sheet. Figure 8 depicts the new schematic. At resonance, the infinite impedance of the AMC makes it possible for one to match the free-space intrinsic impedance, $\eta_{0}$, by simply adjusting the $R_{sheet}$. As such, most of the energy of impinging waves is not reflected by the metasurface but absorbed by the resistive layer.

Another possibility is to exploit the transmission-line equivalence in the design of power-amplifying metasurfaces. Consider an AMC similar to the one of Figure 2, but with slots in the ground plane that couple impinging waves to circuits located behind the metasurface. Figure 9 depicts such a structure, where active circuits can be located in the back of the metasurface. As such, impinging waves can be amplified and then be coupled back to the slots. The final effect is a reflection with power amplification.

**Figure 8 micromachines-14-01405-f008:**
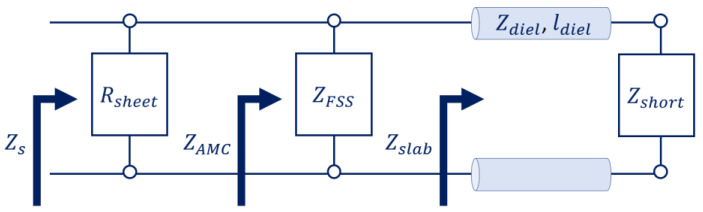
Transmission-line equivalent of the metasurface absorber of [37].

Figure 10 exemplifies one way of modelling the metasurface of Figure 9 with the transmission-line equivalence. In Figure 10, the quadrupole with input and output impedances, $Z_{in}$ and $Z_{out}$, models the scattering parameters of the circuitry behind the ground plane through a matrix, $S_{params}$. Such a matrix takes into account the coupling through the slots, the travelling through transmission lines, the amplification in the active circuits, and so forth. A similar approach was used in [38] to perform an analysis on the challenges related to power-amplifying metasurfaces, including stability issues.

Other types of metasurfaces exist, including transmissive ones, and plenty of methods exist for the modelling thereof. However, in this work, we do not intend to cover all of them. Here, we are mostly interested in metasurface absorbers and power-amplifying metasurfaces, both already exemplified.

### 2.4. Candidate Metasurfaces for Passive Blinking Jamming

We saw in (18) and (19) that one can make $\Phi_{\epsilon }\to \Phi_{\epsilon 1}$ and $\theta_{\epsilon }\to \theta_{\epsilon 1}$ if one can somehow make $S_{\sum 2}\ll S_{\sum 1}$, and $\Phi_{\epsilon }\to \Phi_{\epsilon 2}$ and $\theta_{\epsilon }\to \theta_{\epsilon 2}$ if $S_{\sum 1}\ll S_{\sum 2}$. Furthermore, (5)–(8) indicate that, from the standpoint of platforms targeted by the radar, there are two ways to reduce $S_{\sum 1}$ and $S_{\sum 2}$: the first is to fly away from the radar, increasing the range, $r_{acf}$; the second is to reduce the platform RCS. Hence, the only way to reduce $S_{\sum 1}$ and/or $S_{\sum 2}$ while penetrating in the radar zone is to reduce the platform RCS. Moreover, for blinking jamming purposes, this must be performed in a dynamic manner, for which reconfigurable metasurfaces are essential.

Complex platforms, such as aircraft, drones, and shipment, cause the reflection of waves through diverse mechanisms [39]. To start with, specular reflections happen following Snell’s law due to the impedance discontinuity between the free space and the platform material (usually a metal that can be considered a PEC, whose reflection coefficient is Γ=−1). Moreover, diffraction components are raised by the discontinuities between moving parts, as well as in the tips and corners of the fuselage and embedded weapons [40]. Moreover, the antennas embedded in the platform also contribute to the total reflection through the reradiation of impinging waves. If their impedances are matched with the receivers, conventional antennas will reradiate half the power of the impinging waves. Otherwise, they will reradiate more [41]. In other words, the RCS is the result of reflections and diffractions on diverse parts of the platform, each one contributing more or less to the final radar visibility.

Two mechanisms can be exploited for RCS reduction: the scattering of electromagnetic waves, as this stops them from focusing on the direction of the radar receiver; or the absorption thereof.

Traditionally, the scattering method relied on the adoption of shapes that consisted mainly of faceted flat plates and wedges whose flare angle is less than 180° because the backscatter from this kind of wedge is lower than that of convex surfaces. Furthermore, the design of the platform should orient surfaces so that most of the scattered fields will be directed away from the source of radiation [40]. This method can reduce the monostatic RCS considerably, but the receivers of bistatic or multi-static radars can still be illuminated by the maximum of scattered fields.

The absorption method, however, is based on the matching between the impedances of the free space and the platform material. To this end, special paints based on ferrite or carbonyl iron balls suspended in epoxy resin can be applied over strategic parts of the platform, thereby dissipating the energy of impinging waves through heat [42].

A low RCS can be achieved by combining both the scattering and absorption methods in the project of the aircraft. Good examples of this traditional combination are the Lockheed SR-71 Blackbird [43] and the Lockheed-Martin F-117 Nighthawk [44] aircraft.

In the early 2000s, B. A. Munk disclosed a series of techniques developed for the United States Air Force to reduce the RCS of aircraft. The initial idea was to build radomes using FSSs that allowed the transmission of waves in the antenna operational frequency band but reflected them outside this band. The shape of the radome had to be judiciously chosen such that a specular reflection would not send energy back to the source of radiation [45]. Next, Munk presented a detailed study on the antenna RCS, proposing some methods to reduce it through the impedance matching, which can work well in a narrow bandwidth, or through the choice of proper antenna shapes [46]. The work of Munk can be considered a starting point for what metasurfaces would bring to the RCS reduction technology in the upcoming years.

Metasurfaces can provide a reduction to the RCS through mechanisms similar to the ones commented on above: by scattering, when the spread of energy throughout different directions, polarizations, and frequencies avoids the radar to detect them, or by absorption. A comprehensive review on RCS reduction with metasurfaces is presented in [47].

In the scattering classification, phase-gradient metasurfaces reflect waves to arbitrary directions, allowing the diffusion thereof [48,49,50]. Moreover, checkerboard metasurfaces interchangeably provide out-of-phase reflections to impinging waves, making them cancel each other out in the far field [51,52]. Furthermore, metasurfaces can also reduce the RCS by converting the polarization of an impinging wave to its orthogonal, which can make the radar unable to detect the reflected waves [53,54]. Lastly, time-varying metasurfaces can modulate impinging waves and shift the frequency of the reflected waves, spreading it over the spectrum and making the radar again unable to detect them [55].

The mechanism behind absorptive metasurfaces is the same as in the traditional method: to dissipate the energy of impinging waves through heat. The simplest example, where a resistive sheet added on the top of an AMC does so, is that of [37], already explained with the support of Figure 8. From an aeronautics and space perspective, [37] represented a significant advance in terms of low profile, because the thickness of the metasurface was much smaller than a wavelength, λ. Conversely, traditional Salisbury lens absorbers were about a λ/4-thick [56]. Another way of dissipating energy is through the creation of resonances in lossy dielectrics. In [57], an absorber composed of an electric resonator and a cut wire with reflection and transmission coefficient magnitudes of $\left|\Gamma \right|=0$ and $\left|\tau \right|=0.05$ was proposed. In both [37,57], a narrow bandwidth was expected for the absorption behavior of the metasurfaces because the dissipation mechanism was based on resonances, which is a frequency-dependent phenomenon. One way of circumventing this limitation is to use two or more layers in the metasurface design. For instance, absorption coefficients better than 0.9 were achieved from about 6.0 to 18.0 GHz with a dual-layer metasurface in [20] and from 3.7 to 17.5 GHz with a multilayer metasurface in [21].

The bandwidth limitation is not necessarily a drawback for our blinking jamming application. By exploiting reconfigurability, one can shift back and forth the operational frequency range of the absorber, making it blink from the point of view of a radar tuned at a specific frequency. Many frequency-reconfigurable metasurface absorbers are reported in the literature, for instance, in [58,59,60,61]. In [58], PIN diodes were used as reconfigurable resistors to control a 5 mm thick metasurface absorber. Through biasing the PIN diodes with different voltages, different resistance values are obtained, and the reflectivity characteristics of the metasurface vary mainly in the range from 9.0 to 14.5 GHz [58] (Figure 2c). In [59], reconfigurability was achieved with varactors, whose response can be controlled by different bias voltages. A prototype etched on a FR-4 laminate whose operating frequency could be tuned from 4.3 to 5.9 GHz presented in the measurements an absorption coefficient better than 90%. Out of band, the absorption coefficient remains lower than 10% for high frequencies [59] (Figure 9). The simulated results for diverse angles of incidence are also reported, and the absorption coefficient remains higher than 90% for up to 50° of angle of incidence in both transverse-electric and transverse-magnetic polarizations [59] (Figure 4). The thickness of the metasurface is 1.6 mm. In [60], cuboid ferrite elements were distributed over a ground plane, leading to a dual-band absorptive behavior with a coefficient measured as better than 90% in the lower band and 85% in the higher band. Out of band, the absorption coefficient was measured as lower than 10% for low frequencies [60] (Figure 7). By tuning the ferrite particles with an external magnetic field, the first band is shifted from about 8.7 to 9.7 GHz, whereas the second is shifted from about 9.2 to 10.2 GHz simultaneously. The thickness of the metasurface is 1.8 mm. Ferrite particles were also used in [61], but now with U-shape elements. As such, a quad-band metasurface absorber was designed with an absorption coefficient measured as better than 95% in the lower frequency band and 85% in the others [61] (Figure 8b). Between the bands, the absorption coefficient remains better than 70.0%. With the application of a biasing magnetic field over the ferrite particles, the center frequency of all bands can be tuned from about 8.5 to 10.0 GHz. The thickness of the metasurface is 1.6 mm.

### 2.5. Candidate Metasurfaces for Active Blinking Jamming

Now we turn our attention to metasurfaces that can induce a monopulse radar to estimate the angle $\Phi_{\epsilon }$ as $\Phi_{\epsilon 1}$ or $\Phi_{\epsilon 2}$, and $\theta_{\epsilon }$ as $\theta_{\epsilon 1}$ or $\theta_{\epsilon 2}$, not by alternately reducing the aircraft RCS but increasing it instead. By using power-amplifying metasurfaces, one can also play with the terms of (18) and (19) and make the radar antenna system oscillate.

The application of power-amplifying metasurfaces in blinking jamming is straightforward: by installing these structures over strategic parts of the aircraft, most of the energy of impinging waves can be collected, amplified, and reradiated, which is quite similar to what a DRFM would do, but without a digitizing procedure (which mitigates the problem with spurs) and the need for an antenna and a dedicated compartment.

Plenty of power-amplifying metasurfaces have been reported in the literature [62,63,64,65,66,67,68,69,70,71,72,73,74,75,76,77,78,79]. Back in the 1990s, even before the term ‘metasurface’ was coined, an array of amplifiers surrounded by two FSSs was proposed for an amplification of 11 dB in the transmission of electromagnetic waves at 3.3 GHz [62]. Since then, many other transmissive metasurfaces have been reported in the literature, for example, in [63,64,65]. However, in this work, we are mostly interested in the reflective ones [31,66,67,68,69,70,71,72,73,74,75,76,77,78].

Fundamentally, amplifying reflected waves is a matter of coupling the energy of the waves impinging upon an aperture to a circuitry that can amplify it and couple it back to the aperture for reradiation. Usually, this can be modelled with transmission-line equivalents like the one of Figure 10. In [31,66], slots were etched in the ground plane of an AMC in order to couple the energy of impinging waves in a disposition similar to that illustrated in Figure 9. In those works, the coupled energy was guided through a circuit in the back of the metasurface. There, right-angle transmission lines and amplifiers enabled a polarization-conversion reflection with an amplification of 13.5 dB at 10.0 GHz in [31] (Figure 4c) and 15.93 dB at 5.44 GHz in [66] (Figure 8a). Even though polarization conversion is not interesting for the blinking jamming application, References [31,66] are good examples for the easy understanding of power-amplifying metasurfaces.

The noticeable miniaturization levels achieved by the amplifier technology in the last couple of decades allowed for the design of high-performance circuits whose size is of the subwavelength order in the X-band [67,68] and Ka-band [69]. In the late 2010s, an active circuit topology whose area is $90\times 80\;\mu m^{2}$ was proposed to provide an amplification gain of up to 14 dB at millimeter-wave frequencies as high as 125 GHz [70,71]. Such a circuit was embedded in the unit cells of a 2×2 reflectarray whose gain reached 28 dBi in the measurements.

Recently, amplifiers also started to be extensively applied in reconfigurable intelligent surfaces (RISs). RISs are metasurfaces where the reflection coefficient phase, $\varphi_{\Gamma }$, of each unit cell can be set up in a way that the receiving of signals from specific directions and the reflection thereof towards other directions are both optimized simultaneously. From a communications perspective, the presence of RISs can tailor the wireless propagation channel thereby optimizing the spectral efficiency [72]. This optimization can be performed with the help of artificial intelligence algorithms [73]. To compensate for an indirect propagation path larger than the direct one, a series of works recently proposed the use of amplifiers in the topology of the unit cells of RISs [74,75,76,77]. Most of these works express the performance of the RIS in terms of spectral efficiency, so we avoided using their parameters in our study. One exception is [77], where the authors reported that the amplifiers embedded on the RIS, when biased in the measurements, provided a remarkable gain of 25 dB at 2.36 GHz [77] (Figure 5). Nevertheless, all of these works are important for our research because they validate the feasibility of embedding amplifiers in metasurfaces, reinforcing the trends for a higher technological maturity level soon.

In a recent work [78], a metasurface with a thickness much lower than the wavelength and unit cells composed of cascaded rectangular patches, amplifiers, and phase shifters was proposed for a power-amplifying reflective beamsteering. Such a metasurface, etched on a dielectric laminate Rogers RO4350, can steer the reflected beam to arbitrary directions when illuminated also from arbitrary directions. An experimental gain larger than 25 dB was measured at 5.81 GHz even when the angles of incidence and reflection were as high as 45° and 66°, respectively, and larger than 19 dB for angles as high as 70° and 80° [78] (Table 1). An electronic support (ES) system can provide information about the angle of arrival of the radar signals (similar to what is performed with DRFMs). Then, metasurfaces like the one of [78], embedded in strategic parts of two aircraft, can amplify the reflected signals and increase the aircraft RCS for monostatic radars or even bistatic radars if the location of the radar receiver is somehow also known.

## 3. Results

Aiming to demonstrate the potential of metasurfaces in blinking jamming, an EW scenario simulator is created with MATLAB and the Radar, Antenna, Phased Array System, and Signal Processing toolboxes [79]. A scenario where a dual-aircraft formation is penetrating in the zone of a monopulse radar is simulated. The simulation sampling rate is $f_{s}=100\;MS/s$, and the frequency of carrier is $f_{c}=5.81\;GHz$. The coordinate system of the EW scenario matches the one of Figure 1.

### 3.1. Monopulse Radar and Dual-Aircraft Formation Models

A monopulse radar like the one described in Section 2.1 and operating at $f_{c}=5.81\;GHz$ is located at the position $P\left(0,0,0\right)$. To facilitate the steering of beams during the simulations, each of the four antennas, A–D, is modelled as a phased array composed of 30×30 vertically polarized square-patch elements. To this end, the *patchMicrostrip*, *phased.URA*, *phased.Radiator*, *phased.SteeringVector*, and *phased.PhaseShiftBeamformer* objects are used. The array elements are spaced by λ/2 in a rectangular grid, which leads to a directivity of 34.6 dB to each of the four antennas. As such, the 3 dB beamwidths of each antenna in both main planes are $\Phi_{3dB}=\theta_{3dB}=3.38{}^{\circ}$. The squint angles in relation to the broadside direction, $\theta_{ant}$ and $\Phi_{ant}$, of the antenna system are $\theta_{sqt}=\Phi_{sqt}=3.38{}^{\circ}/4$. Figure 11 depicts the radiation patterns of the antenna system, considering that the antenna system is pointed at the direction $\Phi_{ant}=7{}^{\circ}$ and $\theta_{ant}=12{}^{\circ}$.

To keep the simulation realistic, a gain of 22 dB in the radar transmitter front end is modelled with the *phased.Transmitter* object based on the listed gain of the CA26-4114 amplifier from Ciao Wireless, Inc., Camarillo, CA, USA [80]. The radar receiver, modelled with the *phased.ReceiverPreamp* object, has a gain of 29 dB and a noise figure of 1.3 dB, which are based on the listed features of the CA48-2111 low-noise amplifier, which is also from Ciao Wireless, Inc. [80]. An instantaneous bandwidth of BW=50 MHz and a temperature of 340 K in the radar circuitry (referred to the input) are considered for the computation of noise.

Through its four channels, A–D, the radar transmits identical chirp pulses generated with a *phased.LinearFMWaveform* object with a bandwidth BW=50 MHz and pulse width of PW=8 μs, which leads to a time-bandwidth product of 400. Furthermore, the pulse repetition frequency is 12.5 kHz. For the sake of simplicity, no sort of agility is considered. Hence, the maximum unambiguous range of the radar is 12 km. In the receiving, signals are convoluted with a matched filter modelled with a *phased.MatchedFilter* object.

The desired probability of detection is $P_{d}=0.9$, and false alarm is $P_{fa}=10^{-6}$. The number of non-coherently integrated pulses is 20. Using the Albersheim’s detection equation [81], the minimum required signal-to-noise ratio is calculated as ${SNR}_{min}=2.95\;dB$. The required radar peak power aiming to detect a target whose RCS is $\sigma =1\;m^{2}$ is next calculated as $S_{peak}=1.69\;kW$.

The antenna control system is modelled with (14) and (15) and parameters similar to those used to generate Figure 4: antenna equivalent mass of $m_{ant}=10\;kg$, damping coefficients of $d_{\Phi }=10/s$ and $d_{\theta }=6/s$, and natural frequencies of $f_{\Phi }=4\;Hz$ and $f_{\theta }=2\;Hz$.

A dual-aircraft formation whose centroid is initially located at the position $P\left(r_{acf}=10\;km,\theta_{acf}=15{}^{\circ},\Phi_{acf}=10{}^{\circ}\right)$ is modelled with two *phased.Platform* objects spaced by azimuthal and elevation separations of ∆Φ=1° and ∆θ=0.5°. The formation flies towards the radar at Mach 2 ($v_{1}=v_{2}=686\;m/s$), keeping constant the initial angular separation. The RCS of each aircraft at this orientation is modelled with the *phased.RadarTarget* object as $\sigma =1\;m^{2}$ for the vertical polarization.

The propagation of waves in the scenario is modelled using the *twoRayChannel* object, considering reflections over the sea, whose relative permittivity is assumed to be $\varepsilon_{r}=78.4$. For all the cases hereafter, the simulation starts with the radar antenna system initially pointed at $\theta_{ant}=12{}^{\circ}$ and $\Phi_{ant}=7{}^{\circ}$. Figure 12a depicts this initial scenario by using the *phased.ScenarioViewer* object. The main lobes of each antenna, A–D, are represented by conical lobes whose flare angle is equal to the 3 dB beamwidths, $\Phi_{3dB}$ and $\theta_{3dB}$.

### 3.2. Results with DRFMs

In the first simulation round, the dual-aircraft formation is equipped with DRFMs. For the sake of simplicity, the DRFMs are considered ideal; that is, the retransmitted signals are a perfect copy of the impinging signals, with no delays or spurs. The transmitter, receiver, and antennas are modelled with the same MATLAB objects used in the radar but with different features. To account for a smaller availability of space in the aircraft, an array composed of 10×10 vertically polarized square-patch elements is considered, leading to a directivity of 25.3 dB and 3 dB beamwidths of $\Phi_{3dB}=\theta_{3dB}=10.06{}^{\circ}$. The values of gain for the transmitter and receiver front ends are the same of the radar (22 and 29 dB, respectively), as well as the receiver noise figure (1.3 dB). Furthermore, the jammer peak power is $J_{peak}=158W$, which is enough to perform the blinking jamming in the proposed scenario, as will become clear for the reader in the first simulation. In the matter of noise, we consider that the DRFM is instantaneously receiving signals in a range from 4 to 8 GHz (BW=4 GHz). The DRFM circuitry presents a base temperature of 340 K (referred to the input). The external temperature lows by 2 K for each 1000 ft of altitude, approximately. To account that the circuitry is not in direct contact with the air, 1 K is subtracted for each 1000 ft of altitude in the simulations.

When the simulation starts, the angle errors, $\Phi_{\epsilon }$ and $\theta_{\epsilon }$, are estimated through the aircraft skin returns in the sum and difference virtual channels. Then, these errors are fed to the antenna control system, which, in turn, starts to steer the four beams to new positions closer to the formation. At the instant t=1.5 s, the antenna system points at the centroid of the formation (see Figure 12b), and the jamming is activated. We consider that the jammers have precise information about the radar location provided by an ES system. Hence, their beams are always pointed towards the radar. Figure 13a illustrates the radar antenna angular position, $\Phi_{ant}$ and $\theta_{ant}$, when the blinking frequency is $f_{bl}=f_{\theta }=2\;Hz$.

In Figure 13a, from the instant t=1.5 s, the radar starts to lose stability over the formation, similar to what we saw in Figure 6a. Significant oscillations occur in the elevation dimension, ranging from 14.25° to 15.85°. This range is more than three times the elevation separation of the aircraft ∆θ=0.5°.

Figure 13b shows the system response when the blinking frequency is $f_{bl}=f_{\Phi }=4\;Hz$. This time, oscillations are wider in this dimension, ranging from 7.85° to 12.18°. This range exceeds more than four times the azimuthal separation, ∆Φ=1°. Figure 14 illustrates this scenario at the instants t=2.875 s and t=3.0 s, where we can see a representation of the main lobes of each jamming antenna when they are activated, as well as the deviation of the radar antenna system around the formation.

Last, Figure 13c depicts the system response when the blinking frequency is $f_{bl}=f_{\Phi }$, but now with a jammer peak power reduced by 10 dB, that is, $J_{peak}=15.8W$. The intention is to show the antenna system response when the jamming platform is constrained in available power. We can see that, in this case, small oscillations are present in the radar antenna system which are not enough to avoid the tracking of the formation.

These first simulations serve not only to demonstrate the traditional blinking jamming with DRFMs but also to validate our EW scenario simulator and our monopulse radar model. The obtained results—considering realistic features for the RF components and antennas, the propagation of waves with reflections over the sea, and the presence of noise—are similar to those obtained in Figure 6a,b, where the antenna system response is checked with ideal information about the angular error.

**Figure 14 micromachines-14-01405-f014:**
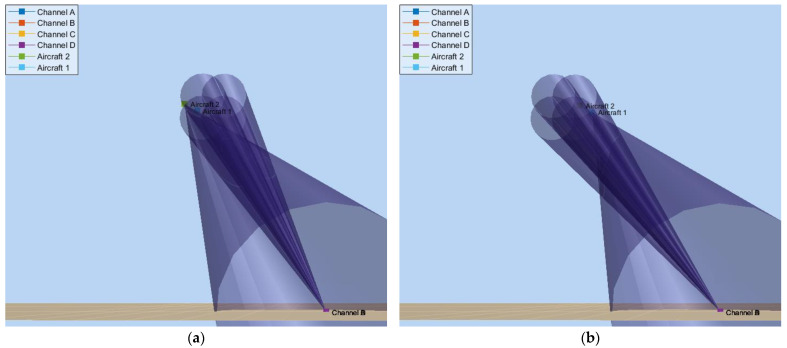
Illustration of EW scenario: (**a**) at the instant t=2.825, the DRFM of Aircraft 2 is activated, attracting the monopulse radar antenna system thereto; (**b**) at the instant t=3.0, the DRFM of Aircraft 1 is activated, calling back the antenna system.

### 3.3. Results for Passive Blinking Jamming with Metasurface Absorbers

Now, we perform simulations of the same scenario, but instead of using DRFMs, we consider that reconfigurable metasurface absorbers are installed over strategic parts of both aircraft in order to reduce the RCS intermittently. As such, a passive blinking jamming, as proposed in Section 2.4, can be performed. For these simulations, we take the results of the metasurface absorber proposed in [59] as a reference for the following reasons: (1) the absorption mechanism is more robust to bistatic and multi-static radars than the scattering mechanism (polarization-conversion metasurfaces could be ineffective against dual-polarized/polarimetric radars, phase-gradient metasurfaces against bi/multi-static radars, and time-varying metasurfaces against wideband radars); (2) the metasurface proposed by [59] presents high absorption coefficients, even for wide angles of incidence; (3) out of band, it presents low absorption coefficients, which is good for the ‘blinking’ of the aircraft RCS when the metasurface is reconfigured in frequency; (4) its low-profile structure is easy to fabricate; (5) the metasurface is etched on FR-4, which is a cheap material; and (6) its tuning mechanism based on varactors is simple.

In this second simulation round, no signals are transmitted from the aircraft. Instead, their RCSs, modelled with the *phased.RadarTarget* object, are varied through two states. First, we consider the ideal case where all the aircraft parts that contribute to the monostatic RCS when the aircraft are flying towards the radar are covered by metasurfaces. Furthermore, the simulated absorption coefficient of 99% seen in [59] (Figure 3) is considered when the blinking jamming is activated. Hence, we vary the aircraft RCS from the state $\sigma_{1}=1\;m^{2}$ to the state $\sigma_{2}=0.01\;m^{2}$. Figure 15a,b depict the radar antenna angular position, $\Phi_{ant}$ and $\theta_{ant}$, when the blinking frequency is $f_{bl}=f_{\theta }=2\;Hz$ and $f_{bl}=f_{\Phi }=4\;Hz$, respectively.

In Figure 15a, the oscillations that occur in the elevation dimension from the instant t=1.5 s range from 14.27° to 15.73°. In azimuth, oscillations range from 9.09° to 10.90°. In Figure 15b, the oscillations in the azimuthal dimension range from 7.91° to 12.09°, whereas those in elevation are in the range 14.90°–15.13°. These oscillations are almost as wide as those of the DRFM case in Figure 13a,b. Hence, at this point, the use of metasurface absorbers for blinking jamming is validated at least from a theoretical point of view.

**Figure 15 micromachines-14-01405-f015:**
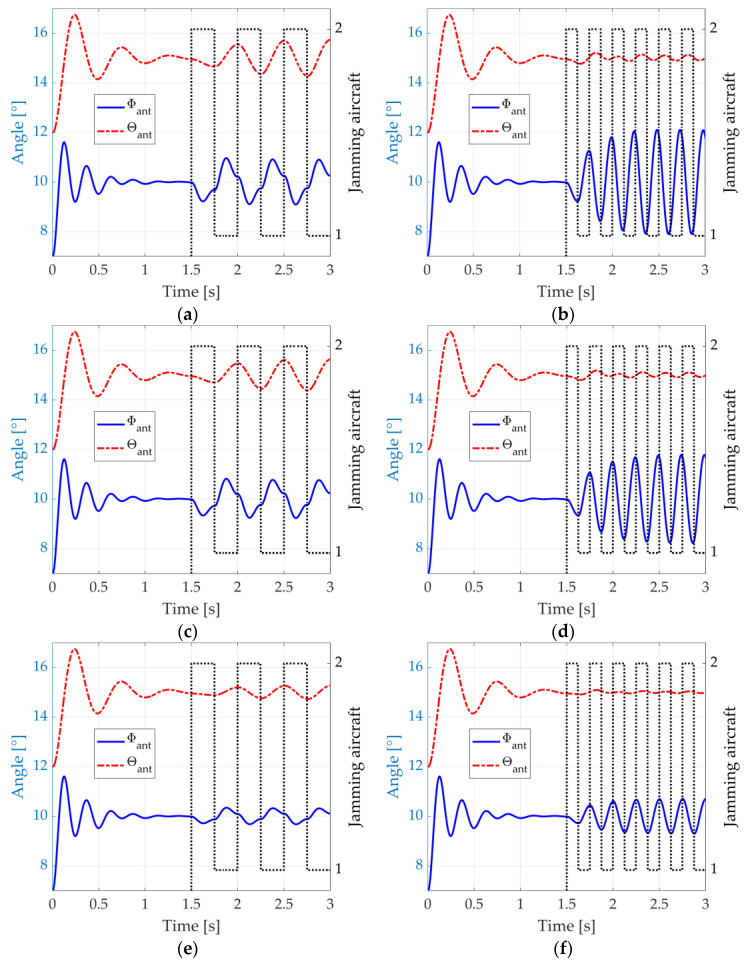
Response of the antenna control system when a passive blinking jamming is performed with metasurface absorbers from t=1.5 s: (**a**) $f_{bl}=f_{\theta }$, $\sigma_{1}=1\;m^{2}$, and $\sigma_{2}=0.01\;m^{2}$; (**b**) $f_{bl}=f_{\Phi }$, $\sigma_{1}=1\;m^{2}$, and $\sigma_{2}=0.01\;m^{2}$; (**c**) $f_{bl}=f_{\theta }$, $\sigma_{1}=1\;m^{2}$, and $\sigma_{2}=0.10\;m^{2}$; (**d**) $f_{bl}=f_{\Phi }$, $\sigma_{1}=1\;m^{2}$, and $\sigma_{2}=0.10\;m^{2}$; (**e**) $f_{bl}=f_{\theta }$, $\sigma_{1}=1\;m^{2}$, and $\sigma_{2}=0.50\;m^{2}$; and (**f**) $f_{bl}=f_{\Phi }$, $\sigma_{1}=1\;m^{2}$, and $\sigma_{2}=0.50\;m^{2}$. The black dotted lines indicate which aircraft is activating its metasurfaces.

The good performance of the metasurface from [59] at oblique angles allows its deployment over multiple parts of the aircraft, even if these parts are oriented in a considerably oblique disposition in relation to the radar. However, the metasurface in Reference [59] is planar, and some curved shapes in the aircraft need conformal structures. Many works exist in the literature to translate planar to conformal metasurfaces, for instance, References [82,83,84,85]. However, we are interested in investigating the practical implementation of blinking jamming when, for some reason, not all the parts of the aircraft that contribute to the monostatic RCS are covered by metasurfaces, or the metasurfaces are not so good in terms of performance. Figure 15c,d show the results for the same cases of Figure 15a,b, but considering that the aircraft RCS can be reduced to $0.10\;m^{2}$ instead of $0.01\;m^{2}$. In Figure 15c, the oscillations in the elevation dimension range from 14.38° to 15.64°, and in the azimuth, they range from 9.23° to 10.77°. In Figure 15d, the oscillations in the azimuthal dimension range from 8.23° to 11.78°, and in elevation, from 14.92° to 15.11°. These results show a neglectable degradation of performance in relation to Figure 15a,b and to the DRFM case, indicating the feasibility of the technique.

We investigated this question further and simulated a third case when only a slight reduction of the aircraft RCS to $0.50\;m^{2}$ can be achieved. The results are shown in Figure 15e,f. The elevation oscillations seen in Figure 15e lie within the range 14.73°–15.28°, and the azimuthal oscillations are in the range 9.68°–10.32°. In Figure 15f, the azimuthal oscillations are within 9.30°–10.70°, and the elevation ones are within 14.96°–15.04°. The aircraft separation is still exceeded in at least one dimension, which is sufficient to make the weapon miss the targets. Therefore, we consider this RCS reduction to be the minimum required to make the passive blinking jamming effective with the use of metasurface absorbers in the proposed scenario.

### 3.4. Results for Active Blinking Jamming with Power-Amplifying Absorbers

Now, we consider that the aircraft have power-amplifier reflective metasurfaces installed over strategic parts thereon. Thus, an active blinking jamming, as proposed in Section 2.5, can be performed. The performance of the metasurface presented in [78], etched on a dielectric laminate Rogers RO4350, is considered in our simulations for the following reasons: (1) This metasurfaces presents a high gain validated by measurements even at wide angles. (2) It presents a simultaneous receiving and reflection beam-steering capability; that is, it can provide high values of reflection gain to directions other than the specular. (3) The low-profile structure is convenient for aeronautics.

In this third simulation round, the RCS of each aircraft is also varied between two states. However, the power-amplifying metasurface increases the RCS, as opposed to the metasurface absorber case, where the RCS is decreased. Initially, we consider again the ideal case where all the aircraft parts that contribute to the monostatic RCS when the aircraft are flying towards the radar are covered by metasurfaces. Then, we simply use the experimental gain measured at 5.81 GHz as larger than 19 dB for incidence angles as high as 70° and transmission angles as high as 80° [78] (Table 1). The wide-angle capability of this metasurface allows one to install them in multiple parts of the aircraft, even if these parts are oriented in a considerably oblique disposition in relation to the radar.

Figure 16a,b show the radar antenna angular position, $\Phi_{ant}$ and $\theta_{ant}$, when the blinking frequency is $f_{bl}=f_{\theta }=2\;Hz$ and $f_{bl}=f_{\Phi }=4\;Hz$, respectively, for two states of RCS: $\sigma_{1}=1\;m^{2}$ and $\sigma_{2}=80\;m^{2}$. The results show angular ranges tightly close to those in Figure 15a,b. The slight differences in both figures are mainly due to noise phenomena.

We repeat the simulations considering metasurface reflection gains of 10 dB and even 3 dB. As such, we consider the cases where not all the parts of the aircraft that contribute to the monostatic RCS are covered by metasurfaces, or where the metasurfaces are not so good in terms of performance. For a reflection gain of 10 dB, we use $\sigma_{1}=1\;m^{2}$ and $\sigma_{2}=10\;m^{2}$, and for a reflection gain of 3 dB, we use $\sigma_{1}=1\;m^{2}$ and $\sigma_{2}=2\;m^{2}$. The results, which are presented in Figure 16c–f, show again angular variations quite similar to those of Figure 15c–f. A significant reason for such similarity is that the active form of blinking jamming using power-amplifying metasurfaces plays with the terms of (18) and (19) exactly as the passive form with metasurface absorbers. Therefore, same ratios between $\sigma_{1}$ and $\sigma_{2}$ result in the same angular variations. One may notice, however, that the jamming oscillations are inverted in both cases, because, in one case, the metasurfaces attract the monopulse radar, whereas, in the other, they repel it.

**Figure 16 micromachines-14-01405-f016:**
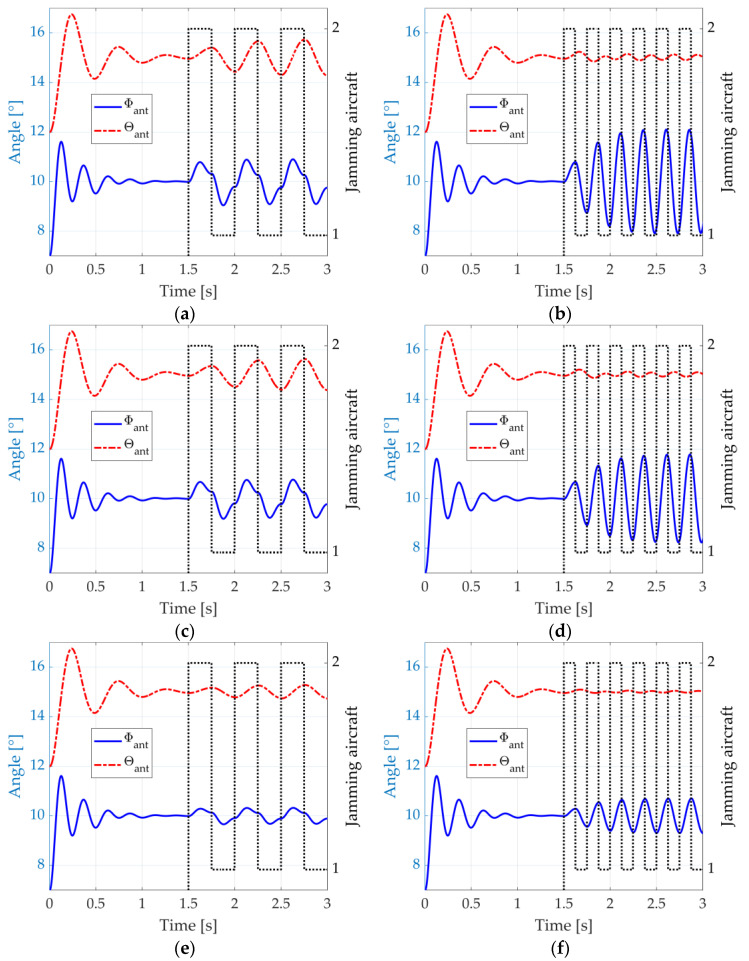
Response of the antenna control system when an active blinking jamming is performed with power-amplifying metasurfaces: (**a**) $f_{bl}=f_{\theta }$, $\sigma_{1}=1\;m^{2}$, and $\sigma_{2}=80\;m^{2}$; (**b**) $f_{bl}=f_{\Phi }$, $\sigma_{1}=1\;m^{2}$, and $\sigma_{2}=80\;m^{2}$; (**c**) $f_{bl}=f_{\theta }$, $\sigma_{1}=1\;m^{2}$, and $\sigma_{2}=10\;m^{2}$; (**d**) $f_{bl}=f_{\Phi }$, $\sigma_{1}=1\;m^{2}$, and $\sigma_{2}=10\;m^{2}$; (**e**) $f_{bl}=f_{\theta }$, $\sigma_{1}=1\;m^{2}$, and $\sigma_{2}=2\;m^{2}$; and (**f**) $f_{bl}=f_{\Phi }$, $\sigma_{1}=1\;m^{2}$, and $\sigma_{2}=2\;m^{2}$. The black dotted lines indicate which aircraft is activating its metasurfaces.

The reflection gain of 3 dB can be considered the minimum RCS ratio required to make the active blinking jamming effective with the use of power-amplifying metasurfaces in the proposed scenario.

## 4. Discussion

The results of the active and passive blinking jamming with metasurfaces are quite similar, with both of them being comparable to the DRFM case even when the RCS variations are not optimal. The main reason for this similarity is that both techniques act over the relationship between the RCS of both aircraft, performing similarly in terms of attracting the monopulse antenna system. Ultimately, both techniques can make the antenna system oscillate in a range wider than the angular separation between the aircraft. Therefore, the main difference between them is essentially in implementation rather than in nature.

The passive blinking jamming uses reconfigurable metasurface absorbers to interchangeably reduce the aircraft RCS, whereas the active blinking jamming employs power-amplifying metasurfaces to increase it. Absorbing electromagnetic waves with metasurfaces is much simpler than detecting the angle of arrival of radar signals with an ES system to amplify them before retransmission in a specific direction. However, ensuring a good ratio between the two RCS states looks easier in the active case, based on amplifiers, than in the passive one. This is because covering most of the parts of the aircraft that contribute to the monostatic RCS when they are flying towards the radar can be a challenge in some aircraft designs. On the one hand, the curved shapes may require conformal metasurfaces, a challenge that can be alleviated with the techniques proposed in [82,83,84,85]. On the other hand, the aircraft surface may be required for other purposes, such as the deployment of sensors.

The fact that blinking jamming requires that the aircraft fly a specific flight path so as to be angularly separated in relation to the target radar may also ease this challenge. Knowing the angular disposition of the problem allows one to plan the deployment of the metasurfaces in the appropriate places.

Another inherent difference between the passive and active techniques is in power consumption. Power-amplifying metasurfaces will consume as much energy as a DRFM when it comes to the retransmission of waves, whereas metasurface absorbers will drain only the necessary power to bias their reconfigurable elements.

A challenge that is common to both passive and active approaches lies in the bandwidth. Their working principle based on resonances makes metasurfaces inherently dependent in frequency. The measurement results of the metasurface of [78] are reported only for 5.81 GHz. Even though the authors of [78] state that no bandwidth limits are inherent to the metasurface they propose, that does not mean that a single metasurface can be used for frequencies far apart in the spectrum. On the contrary, it means that different metasurfaces can be designed to tackle different frequencies. Moreover, the metasurface of [59] can be tuned only from 4.35 to 5.85 GHz, meaning that other versions should be designed to address radars in other frequency ranges. Considering that DRFMs typically can operate in a range of 4 GHz, this feature is a drawback, as it requires that appropriate metasurfaces are set over the aircraft according to specific adversary radars.

All that being said, we state as a future work the design of metasurfaces similar to those of [59,78], used as references in our simulations, but operating in other frequency ranges and also with conformal shapes.

## 5. Conclusions

EW is a highly innovative activity where adversaries have been deploying new technologies to counter one another for many decades. Recent improvements in the technological maturity level of metasurfaces indicate that these artificially engineered structures can be exploited in the EW problem.

This paper proposed the application of metasurfaces in blinking jamming against monopulse radars. After an introductory study on monopulse radars, blinking jamming, and metasurfaces, candidate metasurfaces were seen, and two of them were chosen to serve as references. Their performances were considered in an EW scenario simulator developed with MATLAB and compared with the case of an ideal DRFM. The simulator considers realistic features for the RF components and antennas, the propagation of waves with reflections over the sea, and the presence of noise.

The simulation results show that the implementation of blinking jamming with metasurface absorbers or with power-amplifying metasurfaces is feasible, even if the metasurfaces are not deployed across the entire aircraft.

As opposed to DRFMs, metasurfaces avoid the generation of spurs and the need for a dedicated compartment and antenna. Metasurface absorbers also have the advantage of consuming low amounts of power. As such, the presented techniques pave the way for new ways of performing blinking jamming, such as using numerous small drones to conduct stand-off operations on a large scale in the theater of operation.

Among the challenges to be solved before the technique achieves the theater of operation are the fact that not all the types of metasurfaces already have a conformal version and that the performance thereof is inherently dependent on frequency. The former can be tackled with modern design techniques, but one must know that an off-the-shelf structure will not always be readily available. The latter requires different metasurface versions to tackle different radars operating far apart in the spectrum.

## Figures and Tables

**Figure 1 micromachines-14-01405-f001:**
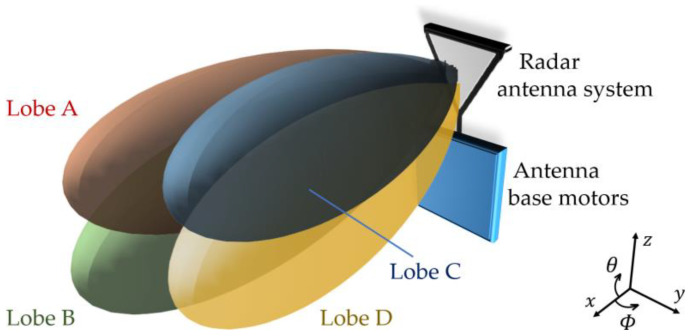
Monopulse radar principle: four independent channels are associated with four feed antennas sharing a same reflector. The coordinate system adopted in this work is indicated.

**Figure 2 micromachines-14-01405-f002:**
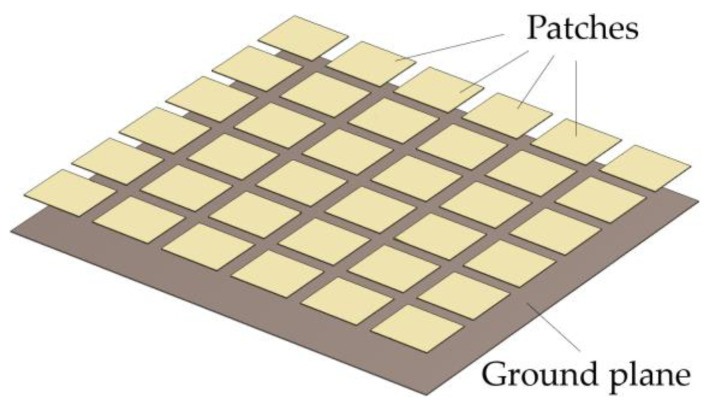
Example of metasurface: AMC without vias.

**Figure 3 micromachines-14-01405-f003:**
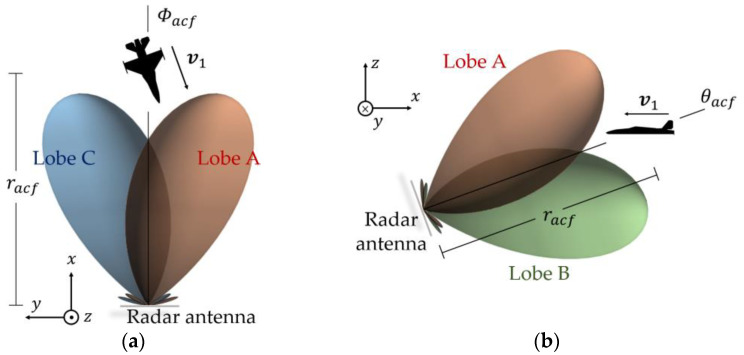
Aircraft illuminated by the monopulse radar: (**a**) azimuth plane and (**b**) elevation plane.

**Figure 4 micromachines-14-01405-f004:**
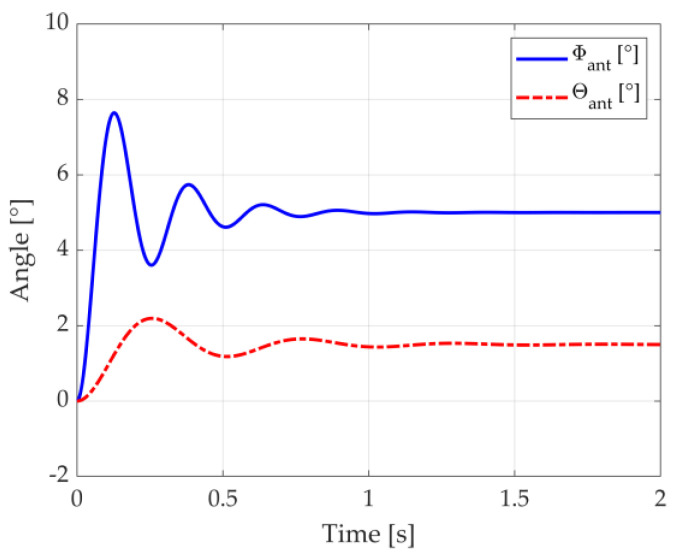
Response of the antenna control system to an aircraft suddenly detected at the angular position $\Phi_{acf}=5{}^{\circ}$ and $\theta_{acf}=1.5{}^{\circ}$.

**Figure 5 micromachines-14-01405-f005:**
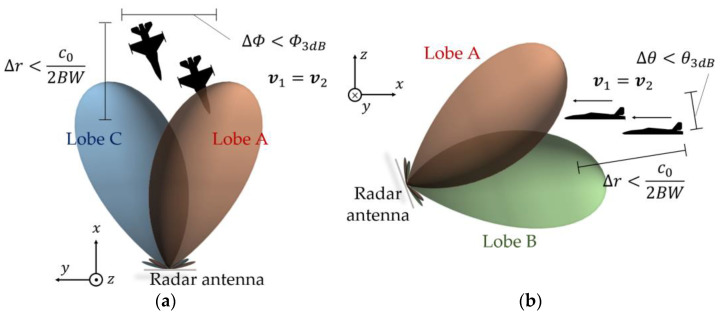
Dual-aircraft formation lying in the same resolution cell of the radar: (**a**) azimuth plane and (**b**) elevation plane.

**Figure 7 micromachines-14-01405-f007:**
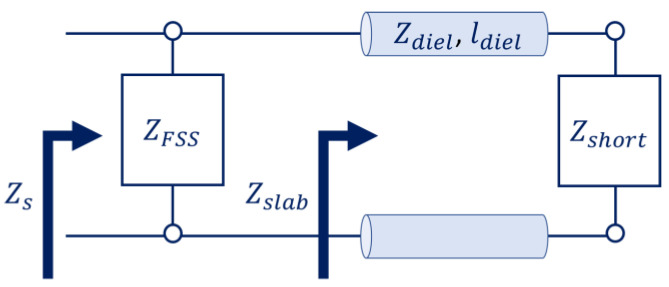
Transmission-line equivalent of the AMC of Figure 2.

**Figure 9 micromachines-14-01405-f009:**
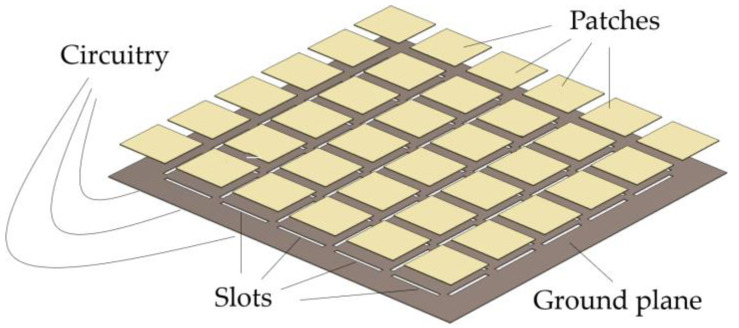
Representation of one type of power-amplifying metasurface.

**Figure 10 micromachines-14-01405-f010:**
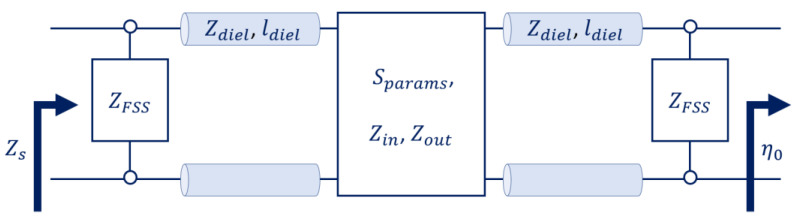
One possibility to model the power-amplifying metasurface of Figure 9.

**Figure 11 micromachines-14-01405-f011:**
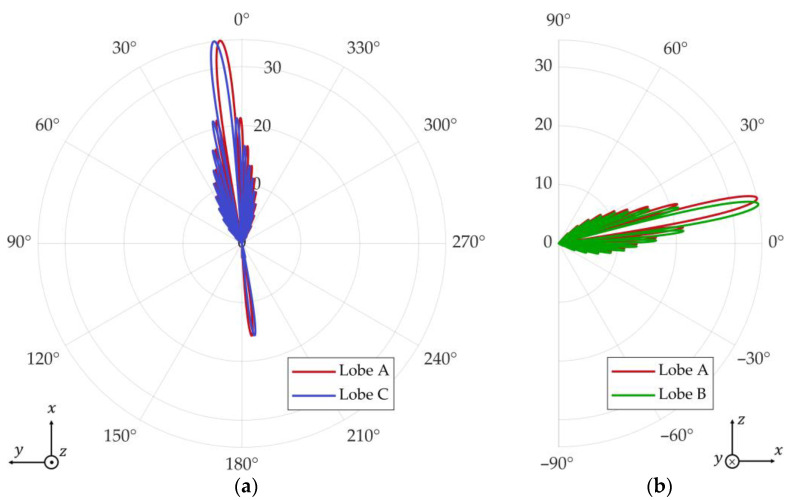
Radiation pattern, in dB, when the antenna system is at the initial position $\Phi_{ant}=7{}^{\circ}$ and $\theta_{ant}=12{}^{\circ}$ : (**a**) azimuth plane and (**b**) elevation plane.

**Figure 12 micromachines-14-01405-f012:**
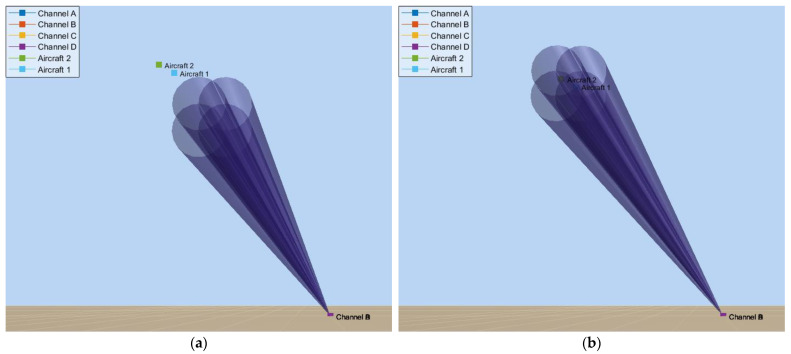
Illustration of EW scenario: (**a**) at the instant t=0, the formation position is $P\left(r_{acf}=10\;km,\theta_{acf}=15{}^{\circ},\Phi_{acf}=10{}^{\circ}\right)$, and the radar antenna direction $\theta_{ant}=12{}^{\circ}$ and $\Phi_{ant}=7{}^{\circ}$; (**b**) at the instant t=1.5 s, $\theta_{ant}≅\theta_{acf}$ and $\Phi_{ant}≅\Phi_{acf}$.

**Figure 13 micromachines-14-01405-f013:**
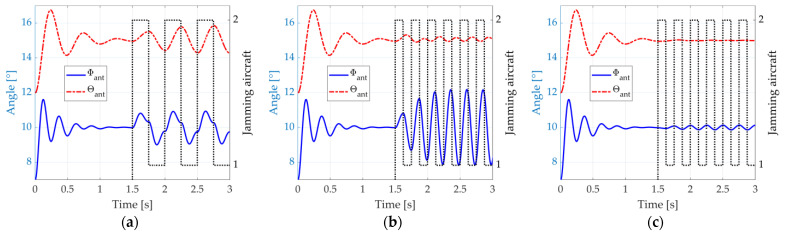
Response of the radar antenna system when blinking-jammed by the dual-aircraft formation equipped with DRFMs in the EW scenario simulator. The jamming starts from t=1.5 s: (**a**) $f_{bl}=f_{\theta }$; (**b**) $f_{bl}=f_{\Phi }$, and (**c**) $f_{bl}=f_{\Phi }$ but with reduced power. The black dotted lines indicate which aircraft is activating its DRFM.

## Data Availability

Data are available upon request.

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
