# Peer review of "Metasurfaces and Blinking Jamming: Convergent Study, Comparative Analysis, and Challenges"

_micromachines, 2023, doi:10.3390/mi14071405_

Round 1
Reviewer 1 Report
The manuscript proposes a reconfigurable metasurface to the implementation of blinking jamming with digital radiofrequency memories. However, there are few comments to be addressed prior to the acceptance at Micromachines.
1. In abstract, Line 9-10, author says current techniques that combined with DRFM are cumbersome and complex... is author’s method ease of implementation? Author is suggested to make a comparison with others in introduction part.
2. In Materials and Methods, materials made of metasurface is missing. Is it easy to fabricate? And how?
Author Response
(A) In abstract, Line 9-10, author says current techniques that combined with DRFM are cumbersome and complex... is author’s method ease of implementation? Author is suggested to make a comparison with others in introduction part.
ANSWER: Thank you for indicating this point, which will improve the clarity of the work. We introduced (highlighted in line 102) an explanation for the implementation of metasurfaces as you suggested:
"Metasurfaces can be implemented by etching, drilling, and deploying lumped elements over dielectric laminates, exactly as done in conventional printed circuit boards".
Now, we believe that it is clear that its implementation is much easier than that of a DRFM, which is a complex system involving many other complex subsystems.
(B) In Materials and Methods, materials made of metasurface is missing. Is it easy to fabricate? And how?
ANSWER: Thank you for pointing that. We introduced the materials of the two metasurfaces chosen for our simulations. In row 617, highlighted in yellow, we have:
"5) the metasurface is etched on FR4, which is a cheap material."
Similarly, in row 662, highlighted in yellow, we have:
"The performance of the metasurface presented in [79], etched on a dielectric laminate Rogers RO4350, is considered in our simulations for the following reasons: "
The question ‘it is easy to fabricate? And how?’ is answered in the point (A) above.
Reviewer 2 Report
The authors present a study on blinking jamming integrated with active metasurfaces, and show the advantages and limitations of the technique with respect to monopulse radars. This is an impressed idea to introduce metasurfaces into applications of blinking jamming. This work is worthy to be recommended for publication in Micromachines, while I would like to hear the responses from the authors related to the following comments:
1)Extended literature review is needed in terms of metasurfaces, especially on the manipulations of electromagnetic wavefronts in the section of Introduction.
2)For more visible and solid effects of the introduction of passive/active metasurfaces, radiation patterns are preferred.
3)How can the authors evaluate the possibilities of spurs from configurable metasurfaces, since basically the considered metasurfaces are electrically tunable. Similar to DRFM, digital circuits are shared.
4)Long sentences should be reworded.
Long sentences should be reworded
Author Response
(1) Extended literature review is needed in terms of metasurfaces, especially on the manipulations of electromagnetic wavefronts in the section of Introduction.
ANSWER: Thank you for indicating this point. In the proposed work, two complex areas converge: blinking jamming and metasurfaces. The current organization of the paper was thoroughly planned to concisely show everything that the reader needs to understand how metasurfaces can be used in the blinking jamming application.
In Section 2.3, a detailed introduction for metasurfaces is presented, starting from the seminal work of Sievenpiper [34], which allows the reader to understand other types of AMCs and, later, absorbers and power-amplifying metasurfaces. Section 2.4 covers the literature review related to phase-gradient metasurfaces, checkerboard metasurfaces, polarization-conversion metasurfaces, time-varying metasurfaces and absorptive metasurfaces (reconfigurable or not). Section 2.5 introduces power-amplifying metasurfaces but really starting from the beginning, when the term metasurface was not coined yet–see [62]. We cover the miniaturization of active circuits. We also approach reconfigurable intelligent surfaces (RIS) in this section before we finally arrive at the power-amplifying metasurface that is used in the simulations.
Hence, we can see that the extended literature review you point out is presented in sections 2.3 through 2.5. Making a literature review of metasurfaces in the introduction would make this section too long. Currently, the introduction already counts three pages. Hence, we will keep the metasurface literature review as it is in the current version of the paper, that is, in sections 2.3 through 2.5.
(2) For more visible and solid effects of the introduction of passive/active metasurfaces, radiation patterns are preferred.
ANSWER: Radiation patterns are a common figure of merit for antennas, but not for metasurfaces. We can see some metasurface works showing radiation patterns when, for instance, the metasurfaces are used to steer beams of electromagnetic waves. However, usually the study of metasurfaces without antennas is conducted through the analysis of their unit cells, which cannot be associated with a radiation pattern. Besides, in our work we are only interested in the RCS values extracted from the metasurfaces that are used in our simulations. Thus, the author does not see a reason for showing a radiation pattern for the metasurfaces in this work.
(3) How can the authors evaluate the possibilities of spurs from configurable metasurfaces, since basically the considered metasurfaces are electrically tunable. Similar to DRFM, digital circuits are shared.
ANSWER: The arise of spurs in DRFMs is associated with the conversion of the signal from the analogue to the digital domain. In DRFMs, the signals are digitized and represented by bits. The number of discretization levels in the digital domain makes the ‘actual’ amplitudes of the signals be represented by approximate values in the digital domain. The error between the actual amplitudes and those in the digital domain raises the spurs.
The fact that the metasurfaces are electrically tunable doesn’t imply that digital circuits are used therein. Indeed, in all the metasurfaces considered in this work, no digital circuits are used. In the passive blinking jamming, the energy of electromagnetic waves is simply absorbed by the metasurfaces and then no spurs exist. In the active blinking jamming, the energy of electromagnetic waves is amplified by analogic circuits embedded in the metasurface unit cells. No digitization is present. Consequently, no spurs are raised.
(4) Long sentences should be reworded.
ANSWER: The work was revised and a limit of three rows for each sentence was checked.